

# Fine-tuning protein language models unlocks the potential of underrepresented viral proteomes

Rajan Sawhney[1], Barbra D. Ferrell[2,3], Thibaut Dejean[1], Zachary Schreiber[2,3], William Harrigan[4], Shawn W. Polson[2,3], K. Eric Wommack[3,5] and Mahdi Belcaid[1]

[1] Department of Information and Computer Sciences, University of Hawaii at Manoa, Honolulu, Hawai'i, United States of America
[2] Department of Computer and Information Sciences, University of Delaware, Newark, DE, United States of America
[3] Delaware Biotechnology Institute, University of Delaware, Newark, DE, United States of America
[4] Hawai'i Institute of Marine Biology, University of Hawaii at Manoa, Honolulu, Hawai'i, United States of America
[5] Department of Plant and Soil Sciences, University of Delaware, Newark, DE, United States of America

## ABSTRACT

Protein language models (pLMs) have revolutionized computational biology by generating rich protein vector representations, or embeddings—enabling major advancements in *de novo* protein design, structure prediction, variant effect analysis, and evolutionary studies. Despite these breakthroughs, current pLMs often exhibit biases against proteins from underrepresented species, with viral proteins being particularly affected, frequently referred to as the "dark matter" of the biological world due to their vast diversity and ubiquity, yet sparse representation in training datasets. Here, we show that fine-tuning pre-trained pLMs on viral protein sequences, using diverse learning frameworks and parameter-efficient strategies, significantly enhances representation quality and improves performance on downstream tasks. To support further research, we provide source code for fine-tuning pLMs and benchmarking embedding quality. By enabling more accurate modeling of viral proteins, our approach advances tools for understanding viral biology, combating emerging infectious diseases, and driving biotechnological innovation.

## INTRODUCTION

Proteins are the molecular machinery of life, serving virtually every essential biological process. Understanding the structural and functional properties of proteins is fundamental to advancing biology and medicine. Artificial intelligence (AI) is profoundly enhancing our insights into protein mechanisms, enabling highly precise predictions of protein structures, interactions, and activities directly from sequence data. Deep learning models, such as AlphaFold (*Senior et al., 2020*; *Jumper et al., 2021*; *Abramson et al., 2024*), have achieved remarkable accuracy in structure prediction, revealing active sites, binding

Corresponding author
Mahdi Belcaid, mahdi@hawaii.edu

pockets, and interaction interfaces essential for understanding protein roles (*Jumper et al., 2021*). Furthermore, AI-driven tools accelerate the discovery of functional motifs and domains, facilitating experimental validation and the design of proteins with customized properties (*Ferruz & Höcker, 2022*). By augmenting empirical methods, AI is transforming proteomics, paving the way for advancements in drug discovery, biotechnology, and synthetic biology (*Qureshi et al., 2023*; *Holzinger et al., 2023*).

Central to these advancements are protein language models (pLMs), which apply principles of large language models to protein sequences. By learning the "grammar" of protein sequences, pLMs capture the complex evolutionary relationships, and functional properties of proteins as presented through structural information, all without explicit targeted training towards any singular goal. At the core of pLM breakthroughs are embeddings—vector representations of protein sequences learned by the model during training. Protein embeddings encapsulate rich, complex biological information (*Lin et al., 2023*), thereby improving performance across bioinformatics applications such as sequence alignment (*Harrigan et al., 2024*; *McWhite, Armour-Garb & Singh, 2023*), function annotation (*Flamholz, Li & Kelly, 2024*; *Hu et al., 2022b*), enzyme classification (*Teukam et al., 2024*), and evolutionary analysis (*Hie et al., 2024*). Additionally, embeddings allow for large-scale analyses that surpass the computational limitations of traditional methods. For example, embedding-based protein similarity replicates evolutionary insights typically derived from computationally expensive phylogenetic approaches while efficiently processing extensive sequence datasets (*Lupo, Sgarbossa & Bitbol, 2022*; *Tule, Foley & Bodén, 2025*).

Various machine learning architectures drive deep learning models of proteins, most notably convolutional neural networks (CNNs) (*Zeng et al., 2016*; *Ragoza et al., 2017*), recurrent neural networks/mLSTM (UniRep, *Alley et al., 2019*), bi-LSTM (*Zhang et al., 2022*), shallow neural networks (using approaches like Word2Vec, *Sharma et al., 2021*), and the currently prevalent Transformer-based architectures (*Vaswani, 2017*). Successful training of pLMs requires vast datasets of protein sequences. Transformers are pre-trained in an unsupervised fashion, with the aim of capturing intrinsic properties of protein sequences which addresses the scarcity of fully annotated protein data. For example, Transformer-based pLMs like ProtBert, ProtT5 (*Elnaggar et al., 2021*), and the ESM family (*Rives et al., 2021*; *Lin et al., 2023*) are trained using a masked language modeling (MLM) objective (*Devlin et al., 2019*) where certain amino acids are masked and predicted based on surrounding context. Other models, such as ProGen (*Madani et al., 2023*; *Nijkamp et al., 2023*), ProtGPT (*Ferruz, Schmidt & Höcker, 2022*), RITA (*Hesslow et al., 2022*) and DARK (*Moffat, Kandathil & Jones, 2022*) use autoregressive causal language modeling approaches for predicting the next amino acid in the sequence (*Bahdanau, 2014*).

Unfortunately, the performance of pLMs and the quality of their generated embeddings varies considerably based on the similarity of the taxonomic levels represented in their training data (*Ding & Steinhardt, 2024*). For example, protein sequences from certain species are assigned higher pLM likelihoods, reflecting biases introduced by the imbalanced representation of species within training datasets (*Ding & Steinhardt, 2024*). This bias can, in turn, adversely affect downstream applications

or render them suboptimal (*Buolamwini & Gebru, 2018*; *Chen, Johansson & Sontag, 2018*; *Kleinberg et al., 2022*; *Shahbazi et al., 2023*). Efforts to mitigate these issues through dataset re-weighting and diversity enhancement have shown promise in improving model performance (*Zhao et al., 2017*; *Ryu, Adam & Mitchell, 2017*; *Yang et al., 2020*; *Gao et al., 2020*; *Rolf et al., 2021*; *Lee et al., 2022*). However, despite these efforts and multi-tiered database sampling strategies (*Elnaggar et al., 2021*; *Rives et al., 2021*; *Lin et al., 2023*), the unbalanced species distribution in protein databases, like UniProt (*The UniProt Consortium, 2024*), often results in similar biases across pLMs. Microbes are particularly impacted by these biases, with many microbial species being underrepresented in large protein datasets (*Ding & Steinhardt, 2024*; *Paez-Espino et al., 2016*; *Youle et al., 2012*; *Mahmoudabadi & Phillips, 2018*). Bacteriophage proteins, such as those targeting ESKAPE pathogens (*Enterococcus faecium, Staphylococcus aureus, Klebsiella pneumoniae, Acinetobacter baumannii, Pseudomonas aeruginosa*, and *Enterobacter* spp.) (*Lee, Hunter & Shim, 2024*), are frequently underrepresented or excluded due to limited annotation (*Sanejouand, 2023*). This lack of representation can limit the ability of pLMs to generalize across diverse microbial taxa, particularly viruses, which make up only a small fraction of public protein databases despite their ubiquity in biological systems (*Moreno-Gallego & Reyes, 2021*).

Fine-tuning—a process of further training pre-trained models on domain-specific datasets—can mitigate biases by refining embeddings to capture diverse sequences and context-specific features. Recent studies suggest this approach has been successful in natural language processing and proteomics (*Ding & Steinhardt, 2024*; *Elnaggar et al., 2021*; *Rao et al., 2019*; *Gu et al., 2021*; *Brandes et al., 2022*; *Rives et al., 2021*; *Howard & Ruder, 2018*; *Devlin et al., 2019*; *Houlsby et al., 2019*). Fine-tuning for enhanced representation learning could potentially be important for viral proteins, where optimization might capture distinct patterns of viral proteins and enhance performance across diverse tasks.

A growing body of work has demonstrated the utility of pLMs in virology, applying them to tasks such as annotating prokaryotic viral proteins (*Flamholz, Biller & Kelly, 2024*), predicting mutations (*Yu et al., 2025*; *Gurev et al., 2025*), and generating viral sequences with specialized models (*Rancati et al., 2024*; *Martin, Gitter & Anantharaman, 2024*; *Marathe, Bajracharya & Yan, 2024*). However, none of these studies have systematically evaluated the impact of fine-tuning general-purpose pLMs to improve performance on viral protein benchmarks, which is the central focus of our study.

A pLM can be fine-tuned by updating all model weights (full fine-tuning), or selectively updating specific parameters of the model. Full fine-tuning models with extremely large parameter counts have large computational demands (100's of GB of RAM), making it infeasible for most standard systems. For example, training the ESM2-15B model took 60 days using over 512 NVIDIA V100 GPUs (*Lin et al., 2023*). However, low-rank adaptation (LoRA) (*Hu et al., 2022a*), a variant of parameter-efficient fine-tuning (PEFT) (*Mangrulkar et al., 2022*), selectively trains specific components of a model, dramatically reducing the number of trainable parameters and computational requirements. Specifically, LoRA decomposes the model weight matrices into a pair of smaller, low-rank matrices, reducing both memory and computational costs. LoRA allows fast, efficient adaptation without

additional inference latency, making it ideal for large models. Using rank r, LoRA adjusts only a small subset of parameters during fine-tuning, mitigating catastrophic forgetting (*French, 1999*; *Kirkpatrick et al., 2017*) and alleviating the RAM burden as pLMs scale in size. A rank of 8, as demonstrated by *Hu et al. (2022a)*, achieves competitive performance when fine-tuning while keeping computational costs low. Higher ranks may lead to better performance in some cases, but they also increase the computational overhead.

This research evaluates the impact of LoRA fine-tuning with three popular representation learning approaches—namely masked language modeling, classification, and contrastive learning—on improving the embedding quality for viral proteins. We compared pre-trained and LoRA-fine-tuned versions of three different and widely popular pLMs: ESM2-3B, ProtT5-XL, and ProGen2-Large. An additional layer with diverse unsupervised or self-supervised learning objectives was uniformly applied across all models, refining learned features to domain-specific applications (*Devlin et al., 2019*; *Raffel et al., 2020*). This standardization enabled fair performance comparisons, minimizing architectural biases (*Radford et al., 2019*; *Elnaggar et al., 2021*).

Our study demonstrates that LoRA fine-tuning for selectively updating pLM parameters using virus-domain specific data and diverse learning objectives enhances downstream bioinformatics performance. All model adaptation and evaluation code is available open-source at GitHub (https://github.com/Hawaii-Bioinformatics/ViralFineTuning) and archived with DOI: 10.5281/zenodo.15865074, offering a practical framework for groups with limited computational resources to leverage pLM adaptation.

## METHODS

### Data

Models were fine-tuned on viral protein sequences from the Virus Orthologous Groups Database (VOGDB) (https://vogdb.org) (*Trgovec-Greif et al., 2024*). VOGDB offers an opportunity for applying both unsupervised and supervised learning approaches by leveraging the orthologous group information provided within the same dataset, facilitating comparisons between pre-trained and fine-tuned models. The VOGDB sequence dataset consisted of 381,760 sequences distributed across 24,916 orthologous groups. The dataset was filtered to only include groups containing over 10 sequences, resulting in 345,261 sequences in 5,981 orthologous groups.

The filtered VOGDB sequence dataset was divided into training and testing subsets in a 90:10 ratio (313,873 training sequences and 31,388 test sequences) using a stratified sampling strategy; all orthologous groups were proportionately represented in the training and test subsets.

For model evaluation, we used four datasets: (1) the VOGDB test subset, generated by selecting 673 VOGs, each containing at least 10 sequences, from the filtered VOGDB sequence dataset. (2) A multiple sequence alignment (MSA) dataset from VOGDB, generated by selecting a subset of 300 VOGs with each VOG's MSA containing 60 to 200 alignments. (3) HeliBase, a dataset of 174 manually curated reference DNA helicase

**Table 1 Protein language models applied in the study.**

| Model | Number of parameters | Max. sequence length | Encoded layers | Embedding size | Database |
|---|---|---|---|---|---|
| ESM2-3B | 3B | 1,024 | 36 | 2,560 | Uniref50/90 |
| ProtT5-XL | 3B | 512 | 24 | 1,024 | BFD100/UniRef50 |
| ProGen2-Large | 2.7B | 1,024 | 32 | 2,560 | UniRef90 |

sequences identified through a combination of historical and recent literature, published metagenomes, and biochemical characterizations (DOI: 10.5281/zenodo.15855844) (*Brosh Jr & Matson, 2020*; *Singleton, Dillingham & Wigley, 2007*; *Nasko et al., 2018*; *Keown, 2024*). (4) The UvsW helicase dataset, containing 47 sequences sourced from Swiss-Prot (*The UniProt Consortium, 2024*). Portions of this text were previously published as part of a preprint (*Sawhney et al., 2025*).

## Model training

Our study used three popular pre-trained pLMs: ESM2-3B, ProtT5-XL and ProGen2-Large (Table 1). For each, we chose a variant with approximately three billion parameters, balancing performance and computational resource requirements, making the model appropriate for a wide range of research environments.

The original pLMs were fine-tuned using three approaches: (1) masked language modeling (MLM), (2) classification with cross-entropy loss (CLS), and (3) contrastive learning with cosine mean squared error loss (CON) (Fig. 1).

**1. Masked language modeling:** Transformer models, such as ESM and ProtT5, leverage MLM (introduced in bidirectional encoder representations from transformers (BERT), *Devlin et al., 2019*) for predicting masked tokens within sequences, and capturing bidirectional context to learn relationships essential for tasks like function prediction and structure analysis. MLM is optimized using cross-entropy loss:

$$Loss = -\sum_{i=1}^{N} \log(P(true\_token_i | context)).$$

Where $N$ is the number of masked tokens, and $P(true\_token_i | context)$ is the probability assigned to the true token at the masked position $i$. Following the approach described in *Devlin et al. (2019)*, we masked 15% of tokens in each protein sequence at random, with masking probability of 80% for ESM2 and 100% for ProtT5 and ProGen2. We computed the loss between the predicted tokens and original tokens using the cross-entropy loss function.

**2. Classification with cross-entropy loss:** Classification-based self-supervised methods are often used due to the availability of large datasets that group proteins into families or orthologous groups. The training was designed to predict an input protein's most likely orthologous group. The model was trained using the cross-entropy loss function, which measured the difference between the predicted probability distribution and the true

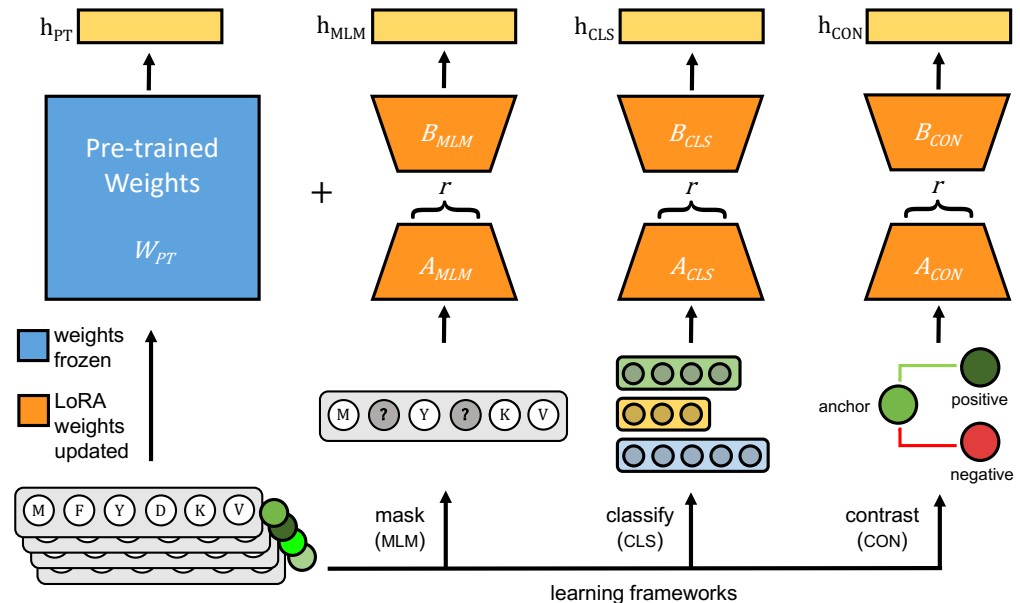

**Figure 1** Each pre-trained protein language model is fine-tuned on viral protein sequences (*x*) using parameter-efficient fine-tuning (PEFT)'s LoRA method. During fine-tuning, pre-trained weights of the pre-trained model ($W_{PT}$), shown in blue, are frozen and only weights for LoRA-trainable parameters, shown in orange, are updated using one of three learning frameworks: masked language modeling (MLM), classification with cross-entropy loss (CLS), and contrastive learning (CON). During inference, $h_{PT}$ is the output of the pre-trained model, and the frozen pre-trained weights and LoRA-learned weights are added together to generate model outputs indicated as $h_{MLM}$, $h_{CLS}$ and $h_{CON}$.

orthologous group label.

$$Loss = -\sum_{i=1}^{N} \log(P(true\_group_i|x_i)).$$

Where $N$ is the number of samples, $true\_group_i$ is the true label, and $P(true\_group_i|x_i)$ is the predicted probability of the true group given input $x_i$.

**3. Contrastive learning:** Contrastive learning is another powerful representation learning method that trains models to embed similar protein sequences closer together in feature space while positioning dissimilar sequences further apart. Siamese networks, a contrastive learning architecture, can be particularly effective for tasks comparing or ranking sequences, such as protein sequence alignment or similarity search. We fine-tuned models with Siamese networks optimized using a hybrid loss function combining cosine similarity and mean squared error (MSE).

$$Loss = \alpha \left(1 - \frac{\sum_{i=1}^{N} A_i B_i}{\sqrt{\sum_{i=1}^{N} A_i^2} \cdot \sqrt{\sum_{i=1}^{N} B_i^2}}\right) + (1-\alpha) \cdot \frac{1}{N} \sum_{i=1}^{N} (A_i - B_i)^2.$$

Where $A$ and $B$ are the two vectors being compared, $N$ is the dimensionality of the vectors, and $\alpha$ is a weight factor balancing the cosine and MSE loss. The Siamese network was trained on triplets of protein sequences: an anchor, a positive (a sequence from the same

**Table 2  Fine-tuning specifications for the protein language models.**

| Model | Storage size (MB) | Training with Adam (GB) | Trained parameters (LoRA) | Training time (hrs) | Train loss | Test loss | Hardware (GPU) |
|---|---|---|---|---|---|---|---|
| ESM2-3B$_{MLM}$ | 51 | 42.34 | 4.42M | 127 | 0.048 | 0.051 | 1 Nvidia A6000 |
| ESM2-3B$_{CLS}$ | 52 | 42.34 | 4.42M | 146 | 6.424 | 8.731 | 1 Nvidia A6000 |
| ESM2-3B$_{CON}$ | 62 | 51.48 | 4.99M | 150 | 0.063 | 0.085 | 2 Nvidia A6000 |
| ProtT5-XL$_{MLM}$ | 97 | 24.60 | 8.85M | 48 | 0.011 | 0.124 | 4 Nvidia A100 |
| ProtT5-XL$_{CLS}$ | 97 | 34.50 | 8.85M | 136 | 5.283 | 8.624 | 1 Nvidia A6000 |
| ProtT5-XL$_{CON}$ | 48 | 39.40 | 4.09M | 150 | 0.117 | 0.127 | 1 Nvidia A6000 |
| ProGen2-Large$_{MLM}$ | 46 | 22.40 | 3.93M | 100 | 0.862 | 0.874 | 1 Nvidia A6000 |
| ProGen2-Large$_{CLS}$ | 46 | 23.10 | 3.93M | 100 | 5.321 | 8.813 | 1 Nvidia A6000 |
| ProGen2-Large$_{CON}$ | 48 | 25.40 | 4.09M | 120 | 0.864 | 0.961 | 1 Nvidia A6000 |

**Notes.**
Appended subscripts indicate the model's training type: masked language modeling (MLM), classification (CLS), or contrastive (CON).

orthologous group), and a negative (a sequence from a different orthologous group) according to the method described in *Schroff, Kalenichenko & Philbin (2015)*. We used the cosine MSE loss for computing the similarity between the embeddings. Specifically, the cosine similarity was calculated for the anchor–positive and anchor–negative pairs, and the MSE was applied to encourage anchor–positive similarity to be close to 1 and anchor–negative similarity to approach 0.

These methods can also be integrated. For example, XLNet (*Yang et al., 2019*) combined autoregressive objectives while adhering to the contrastive learning framework. Through these representation learning methods, the model captures complex biological relationships, performing optimally on downstream tasks relevant to viral protein function and mutation analysis.

## Training and implementation specifications

All models were implemented in PyTorch (v.2.4.0), using HuggingFace implementations of ESM2 (facebook/esm2_t36_3B_UR50D) and ProtT5 (Rostlab/prot_t5_xl_uniref50) from the transformers package (v.4.42.4), ProGen2 from its github repository (https://github.com/enijkamp/progen2), and LoRA from the peft package (v.0.10.0) (https://pypi.org/project/peft). Models were trained using an NVIDIA A6000 graphics card with 48GB memory and an NVIDIA A100 with 80GB memory. Model weights were optimized *via* backpropagation using the Adam optimizer (*Kingma & Ba, 2014*) with a learning rate of $1e-9$ for epochs ranging from 2 to 10. We used LoRA $r = 8$ to fine-tune the entire updated architecture (pLM and PEFT layers). LoRA matrices were added to query, key, and value weights (*Wq, Wk,* and *Wv,* respectively) and learned through the variant's newly incorporated representation learning frameworks. Model parallelism was used for ESM2-3B given its memory demands with parameters assigned to designated GPUs. Table 2 provides model training specifications and their average loss values.

## Protein embeddings

Protein embeddings were generated by tokenizing each residue in the protein sequence and using the weights from the model's last hidden layer to extract a vector for each

token. Each fine-tuned model's LoRA-learned weights are added to the pre-trained weights for extracting residue vector representations. The dimensionality of these vectors was defined by the model's embedding size (Table 1). Padding and special tokens were removed for matching the number of vectors to the residues. Sequence-level embeddings, or pooled embeddings, were obtained by averaging the amino acid embeddings across the sequence. All test sequences were embedded prior to evaluation, except for vector clustering alignments (vcMSA, *McWhite, Armour-Garb & Singh, 2023*) alignments, where embeddings were generated on the fly.

## Model evaluation

Our model evaluation presumed that improvements in protein embedding quality will be reflected in both statistical evaluations and downstream tasks. Specifically, we assessed the embeddings generated from each pre-trained and fine-tuned model using pairwise comparisons, clustering, and alignment-based approaches to determine whether fine-tuning improved the overall quality of the embeddings.

### *Experiment 1: Pairwise comparison-based experiments*

We hypothesized that fine-tuning results in high quality embeddings that lead to more accurate and effective detection of similarity when compared with the pre-trained model. This hypothesis was tested using three pairwise comparisons: (1) sequence similarity across proteins within the same orthologous group; (2) similarity at conserved positions in MSAs; and (3) similarity at non-homologous positions in MSAs. All three experiments utilized cosine similarity, a robust metric for comparing embeddings based on vector orientation, which avoided biases related to sequence length and composition.

## Pairwise sequence comparisons with pooled embeddings

To evaluate whether fine-tuning improved the quality of pooled embeddings, we computed pairwise cosine similarity between the 10 members of each of 673 VOGs in the VOGDB test subset (30,285 within-family pairwise comparisons).

## Similarity within conserved MSA columns

To investigate conserved and non-conserved MSA positions, from the VOGDB MSA dataset, we identified 300 VOGs, with each VOG containing between 60 and 200 alignments. Conserved positions were defined as those with fewer than 5% unknown residues (denoted as "X" according to IUPAC standards), consisting of at most four different amino acid types, and with at least 70% of the residues in the column composed of the most abundant amino acid. Each MSA could contain multiple conserved columns. For each conserved column, we calculated the pairwise amino acids embedding cosine similarity. We limited our analysis to approximately 1 million pairwise conserved site comparisons for tractability, resulting in 1,145,044 comparisons across 193 conserved columns covering 5,323 sequences from 51 VOGs.

## Similarity across random sites

The ability of fine-tuned models to discriminate between non-homologous positions across sequences, was evaluated using an iterative sampling method for sequences within an MSA.

For each test iteration, embeddings of amino acid pairs derived from distinct positions and sequences were compared, ensuring that these amino acids did not originate from conserved columns and were separated by at least 20 positions. More formally, for each iteration, a random position $p_i$ is selected for each sequence $s_x$ and $p_j$ for each sequence $s_y$, such that $p_i \neq p_j$ and $|p_i - p_j| > 20$ for all $x \neq y$. This sampling methodology ensured that embeddings were compared between non-homologous, well-separated positions across distinct sequences. This procedure yielded a total of 1,049,148 pairwise cosine similarity comparisons across 19,071 sequences from 196 VOGs.

### Experiment 2. Clustering-based experiments

An iterative clustering analysis using pooled sequence embeddings was used for determining whether fine-tuning produces more distinct and biologically meaningful clusters compared to pre-trained models. To ensure robust and comparable estimates, we ran 1,000 bootstrap iterations. In each iteration, 500 VOGs were randomly sampled from the VOGDB test subset, including all their corresponding sequences (at least 10 per group). This sampling strategy fixed the number of groups per replicate, preventing a few very large VOGs from dominating the silhouette score and providing 1,000 independent estimates for variance assessment. Clustering was performed using the Hierarchical Density-Based Spatial Clustering of Applications with Noise (HDBSCAN) algorithm (*Campello, Moulavi & Sander, 2013*; *McInnes, Healy & Astels, 2017*) on the pooled sequence embeddings from each sampled subset. Cluster quality was assessed by computing their silhouette scores, providing a quantitative measure of distinctiveness and cohesion. The highest silhouette score achieved across all iterations for each model served as a complementary "best-case" metric, highlighting each model's maximal ability to distinguish biologically meaningful patterns among viral proteins.

HDBSCAN parameters for each model suite were optimized individually to attain the best silhouette scores. For the ESM2-3B and ProtT5-XL models, the parameters were set to min_cluster_size=10, min_samples=5, epsilon=1.0, and alpha=1.0. In contrast, ProGen2-Large required a min_cluster_size of 20 to avoid consistent negative silhouette scores. The silhouette score was computed as:

$$S(i) = \frac{(b(i) - a(i))}{max(a(i), b(i))},$$

where a(i) is the mean intra-cluster distance and b(i) is the mean nearest-cluster distance. The overall silhouette score, computed as the mean of S(i) across all points, ranges from −1 to 1, with higher values indicating superior clustering.

Further, the ability of fine-tuned models to improve protein family and function identification was assessed through clustering of diverse DNA helicase sequences within HeliBase. Given the limitations of traditional homology-based methods in precise family-level functional annotation (*Flamholz, Li & Kelly, 2024*), helicases served as an effective candidate for evaluating model's efficacy in achieving precise family-level assignments.

Helicase embeddings derived from pre-trained and fine-tuned models were converted to two-dimensional vectors using Uniform Manifold Approximation and Projection for Dimension Reduction (UMAP, *Sainburg, McInnes & Gentner, 2021*). HDBSCAN

(min_samples=5, min_cluster_size=5) was used for clustering UMAP representations. To compare the performance of embeddings with conventional phylogenetic approaches, a phylogenetic tree was constructed alongside Pfam domain analysis of all helicases. Using InterProScan (version 5.67-990) (*Jones et al., 2014*) and the Pfam database, significant domain hits, with start and stop regions were obtained. Full length helicase sequences were aligned with MAFFT (–auto mode, version 7) (*Katoh & Standley, 2013*), and an approximate maximum-likelihood tree was produced using FastTree (version 2.1) (*Price, Dehal & Arkin, 2010*) and annotated with domain information in Iroki (*Moore et al., 2020*).

### *Experiment 3: Alignment-based experiments*

These experiments evaluated whether fine-tuning improved the quality of protein embeddings specifically for pairwise soft alignments (*Harrigan et al., 2024*) and vector-clustering MSAs using vcMSA (*McWhite, Armour-Garb & Singh, 2023*). It was expected that fine-tuned embeddings would yield longer soft alignments of homologous residues and improved vcMSA consistency as compared with pre-trained models. Such an outcome would demonstrate a model's enhanced ability for capturing biologically meaningful relationships within and between protein sequences.

## Soft alignments

Pairwise soft alignments were generated using amino acid-level embeddings for sequence pairs belonging to the same VOG, derived from the VOGDB test subset (resulting in a total of 30,285 pairwise soft alignments) (*Harrigan et al., 2024*). Resulting pairwise alignments were filtered to retain only those exhibiting path length (longest homologous residue path) exceeding 18 for sequences shorter than 1,024 residues for all models. We used the length of the soft alignment path as a metric for comparing the quality of embeddings produced by each model, with the expectation that fine-tuned embeddings would result in longer paths of protein sequences exhibiting remote homology.

## Multiple sequence alignments

MSAs were generated using the vcMSA algorithm (*McWhite, Armour-Garb & Singh, 2023*) on amino acid-level embeddings of sequences from 100 randomly sampled VOGs from the VOGDB test subset. A maximum of 150 sequences were randomly selected from these VOGs and were shuffled, increasing alignment difficulty and simulating realistic alignment challenges. The vcMSA algorithm, which originally used the pre-trained ProtT5 model for generating amino acid-level embeddings, was modified to incorporate all models developed in this study. Fine-tuning should enhance the accuracy of vcMSA alignments by refining the contextual embeddings used for sequence clustering and ordering, thus yielding alignments with greater biological relevance compared to those derived from pre-trained embeddings. The generated vcMSAs were evaluated using T-Coffee's (*Notredame, Higgins & Heringa, 2000*) Transitive Consistency Score (TCS) (*Chang, Di Tommaso & Notredame, 2014*), a metric that measures the reliability and consistency of aligned sequences across various alternative alignments.

Further, as a focused analysis of UvsW helicase proteins from bacteriophages, we evaluated the quality of the vcMSA using mutual information (MI) and occupancy,

comparing against conventional alignment methods (EMBOSS Needle pairwise alignments and Clustal Omega MSAs, *Madeira et al. 2024*). MI quantifies the dependency between positions in an MSA, indicating co-evolutionary relationships, while occupancy measures the proportion of sequences with residues (non-gaps) at a position, reflecting conservation and alignment completeness.

## RESULTS

Overall, pLMs ESM2-3B, ProtT5-XL, and ProGen2-Large benefited from fine-tuning with viral sequences based on statistical analyses and the outcome of downstream tasks performed with sequence embeddings. Fine-tuning improved the overall embedding quality based on: (1) pairwise comparisons evaluating pooled embeddings and sequence similarity within conserved or non-homologous MSA positions; (2) clustering; and (3) alignments.

### Experiment 1: Pairwise comparison-based experiments

Fine-tuning should improve embedding quality, resulting in increased cosine similarity between similar proteins or amino acids, and decreased cosine similarity between non-homologous proteins or amino acids. Evaluation of model performance was based on three pairwise embedding assessments: (1) pooled embedding similarity for proteins within the same VOG (30,285 comparisons across 673 groups in the VOGDB test subset); (2) amino acid-level embedding similarity at conserved MSA positions (1,145,044 comparisons across 193 conserved sites over 5,323 alignments from 51 orthologous groups); and (3) amino acid-level embedding similarity at non-homologous MSA positions (1,049,148 comparisons across 19,071 alignments from 196 orthologous groups).

Overall, fine-tuned versions of the three models outperformed their original pre-trained model, though the scale of improvement depended on the training framework. Notably, fine-tuning of the ESM2-3B pLM using the contrastive model demonstrated the most significant improvement. The pre-trained ESM2-3B model yielded consistently high cosine similarities across all tests (Fig. 2), which aligns with a known limitation wherein embeddings using this pLM yield a narrow distribution of high cosine similarity scores, regardless of the evolutionary or functional relationships between the pairs of proteins compared (*Tran, Khadkikar & Porollo, 2023*). Fine-tuning broadened this distribution, particularly in distinguishing conserved from non-conserved sites.

As illustrated in Fig. 2A, the pre-trained ESM2-3B model exhibited a median cosine similarity of 0.99 for pooled embeddings, even though the average pairwise protein similarity (global alignment using BLOSUM62 scoring matrix) within individual protein families was only 0.51 (see Supplementary Material, section 1.2). The narrow cosine similarity distribution observed in pre-trained ESM2-3B broadens significantly after fine-tuning. The impact of fine-tuning was particularly evident when comparing the cosine similarity of amino acid embeddings at conserved *versus* random sites (Figs. 2B and 2C). Fine-tuning ESM2-3B using the contrastive model exhibited the largest cosine similarity difference between conserved and non-conserved sites (0.14). This large difference reflected better performance in distinguishing conserved from non-conserved sites. All other models

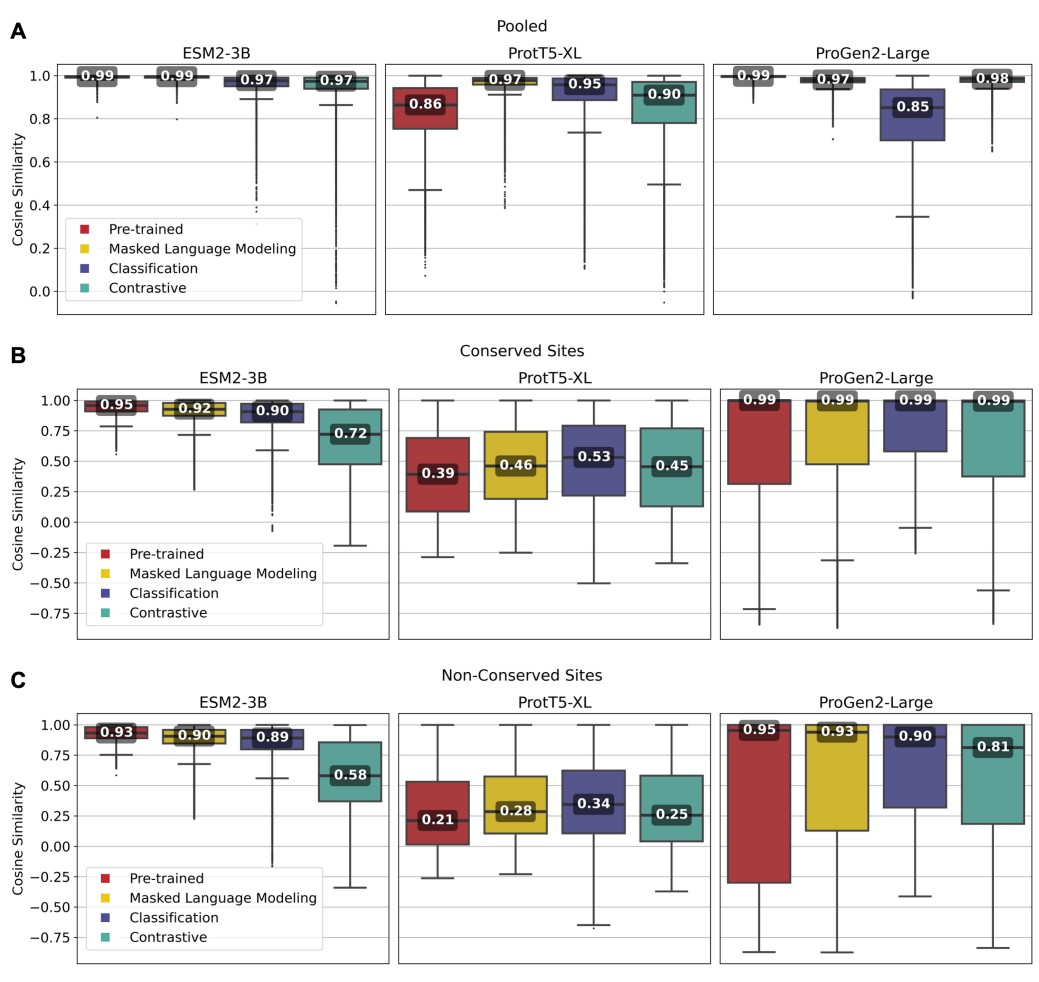

**Figure 2** Pairwise cosine similarity distribution and median values for (A) pooled sequence embeddings of sequences within an orthologous group, (B) conserved sites within multiple sequence alignments, and (C) non-conserved sites within multiple sequence alignments.

and the pre-trained model showed little difference between conserved and non-conserved sites.

Fine-tuning of the ProtT5-XL pLM resulted in only marginal improvements (Fig. 2, middle column). Across all pairwise comparisons, the changes in cosine similarity scores for all fine-tuned ProtT5-XL models were minimal compared to the pre-trained model.

Similarly, fine-tuning of Progen2-Large did little to improve performance according to cosine similarity. When looking at non-conserved regions (Fig. 2C), the median scores remained high, suggesting that fine-tuning did not enhance the model's ability to differentiate between homologous and non-homologous amino acids.

## Experiment 2: Clustering-based experiments

Pooled embeddings of sequences from randomly selected 500 VOGs from the VOGDB test subset were clustered using HDBSCAN, and cluster quality was evaluated with silhouette scores for assessment of whether fine-tuning improved formation of biologically meaningful

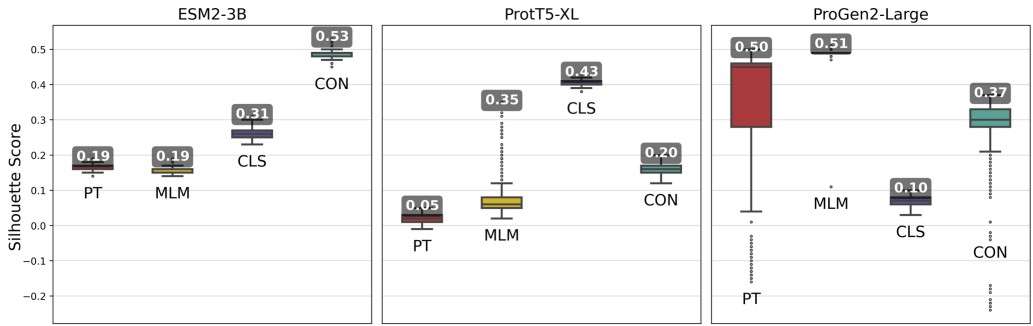

**Figure 3** **Silhouette scores of embedding clusters across 1,000 iterations.** Each iteration used Hierarchical Density-Based Spatial Clustering of Applications with Noise (HDBSCAN) to generate clusters of sequence pooled embeddings from 500 randomly selected VOGs from the VOGDB test subset. Box plots represent the silhouette score distribution across all iterations; numbers correspond to the maximum silhouette score observed, illustrating each pLM's best-case performance. ESM2-3B and ProtT5-XL results with min_cluster_size=10; Progen2-Large results with min_cluster_size=20; pre-trained (PT), masked language model (MLM), classification (CLS), or contrastive (CON) indicate the model's training type.

clusters. This process was repeated over 1,000 iterations, with the maximum scores used to compare each models' capability for capturing meaningful distinctions among viral proteins.

Fine-tuned ESM2-3B and ProtT5-XL models achieved the same or higher silhouette scores (Fig. 3) compared with their pre-trained pLMs. Notably, contrastive fine-tuning of ESM2-3B showed a score approximately three times higher than that of its pre-trained model. Among the ProtT5-XL fine-tuning models, the classification approach yielded the highest silhouette score, but still underperformed compared to contrastive fine-tuning of ESM2-3B.

Fine-tuning approaches with the ProGen2-Large model produced widely varying changes in silhouette scores. By and large, fine-tuning approaches performed worse than the pre-trained model, indicating that additional training led to limited gains. Interestingly, selecting cluster size of minimum size 10, while optimal for the ESM2-3B and ProtT5-XL models, consistently resulted in negative silhouette scores for ProGen2-Large embeddings. Thus, increasing the minimum cluster size to 20 was necessary for this pLM.

The ESM2-3B pLM, fine-tuned using the contrastive approach, was chosen for further performance evaluation based on observations of the cosine similarity (Fig. 2) and silhouette score (Fig. 3) tests. Using the contrastive ESM2-3B model we assessed whether fine-tuning improved meaningful cluster formation for well known proteins having extensive family and functional domain annotations. Comparison of HDBSCAN cluster distributions from the pre-trained (Fig. 4A) and contrastive fine-tuned ESM2-3B pLMs (Fig. 4B) for DNA helicase sequences within HeliBase demonstrated dramatic improvements in groupings according to helicase families. Member proteins within helicase families SF1 and SF2 were clearly separated using embeddings generated with the contrastive fine-tuned ESM2-3B whereas embeddings generated with the pre-trained ESM2-3B mostly overlapped for these families. Interestingly, the contrastive ESM2-3B differentiated two subgroups within

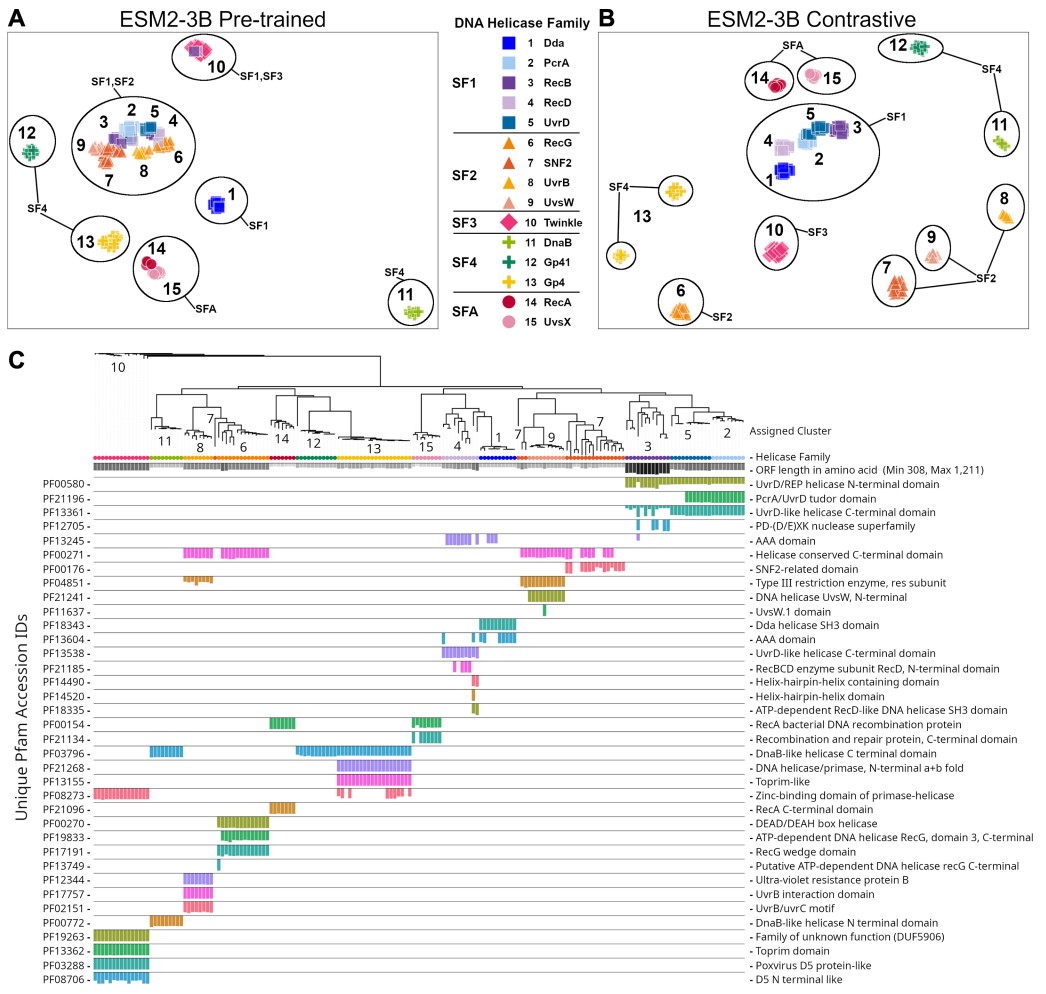

**Figure 4** **Family-level clusters of 174 DNA helicase sequences using embeddings from (A) pre-trained and (B) contrastive ESM2-3B models.** Applying Hierarchical Density-Based Spatial Clustering of Applications with Noise (HDBSCAN) to (A) ESM2-3B pre-trained embeddings led to limited separation among helicase families, while (B) contrastive ESM2-3B embeddings led to clearer distinctions. (C) FastTree approximate maximum-likelihood tree of full length helicase sequences, along with their domain composition identified through InterProScan. Clades in the tree are annotated with HDBSCAN cluster numbers, and leaf nodes are colored to correspond to their respective helicase families from (A) and (B) using Iroki. Bars below the tree indicate relative domain coverage, labeled by Pfam identifier (left) and domain annotation (right).

the Gp4 family (cluster 13 in Fig. 4B) based on the strict presence or absence of a Pfam PF08273 zinc-binding domain, a distinction missed in more classical phylogenetic tree analysis (yellow clade, Fig. 4C).

## Experiment 3: Alignment-based experiments

We evaluated whether fine-tuning improves protein embeddings for alignment-based tasks, including soft alignments and vcMSAs. Fine-tuned models were hypothesized to improve the identification of homologous residues, yielding longer alignments and enhancing

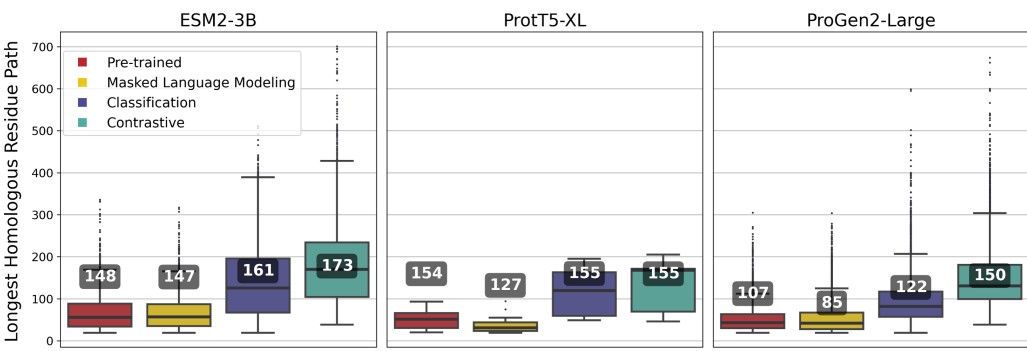

**Figure 5** **Longest homologous residue paths identified from pairwise soft alignments where the contrastive variant produced paths at least twice as long as those identified by the pre-trained model.** Box plots represent this subset; numbers correspond to the overall mean of path lengths across all 27,046 pairwise alignments.

vcMSA quality, thereby reflecting an improved ability to capture biologically meaningful evolutionary relationships.

## Soft alignments

Pairwise soft alignments were generated from amino acid-level embeddings for 30,285 sequence pairs belonging to the same VOG in the VOGDB test subset (*Harrigan et al., 2024*). Only those pairwise alignments with a path length longer than 18 for sequences shorter than 1,024 residues were retained for model assessment. Fine-tuned models were expected to produce longer paths of homologous residues.

ESM2-3B fine-tuned using the contrastive approach outperformed the pre-trained and other fine-tuned models, achieving a mean homologous residue path length of 173 (Fig. 5). However, mean values alone offer only a partial view of the improvements. Among 27,046 alignments, 1,665 saw the residue path length double, indicating that contrastive fine-tuning substantially enhanced the quality of alignments generated using ESM2-3B embeddings. The box plots in Fig. 5 depict the distribution of path lengths for this subset of alignments—those where contrastive fine-tuning at least doubled the path length relative to the pre-trained model. In contrast, the annotated mean values were computed across all pairwise alignments in the test set. Contrastive fine-tuning also improved the ProtT5-XL and ProGen2-Large models, yet neither of these improved models demonstrated as substantial an improvement of path length as the contrastive ESM2-3B model.

Soft alignment using two Megamimivirinae putative ankyrin repeat protein sequences: AFX93168.1 from Megavirus courdo11 (*Megavirus; Megavirus chilense*) and AHA45685.1 from Hirudovirus strain Sangsue (*Mimivirus*) illustrate improvements in sequence alignment based on contrastive fine-tuning of ESM2-3B. The pre-trained ESM2-3B model identified a homologous residue path of 71 residues for this sequence pair (Fig. 6A), whereas the contrastive fine-tuned model significantly extended this path to 174 residues (Fig. 6B). A BLASTP pairwise sequence alignment produced a highly fragmented result comprising 43 high-scoring pairs with an *E*-value of 0.089 (Fig. 6C).

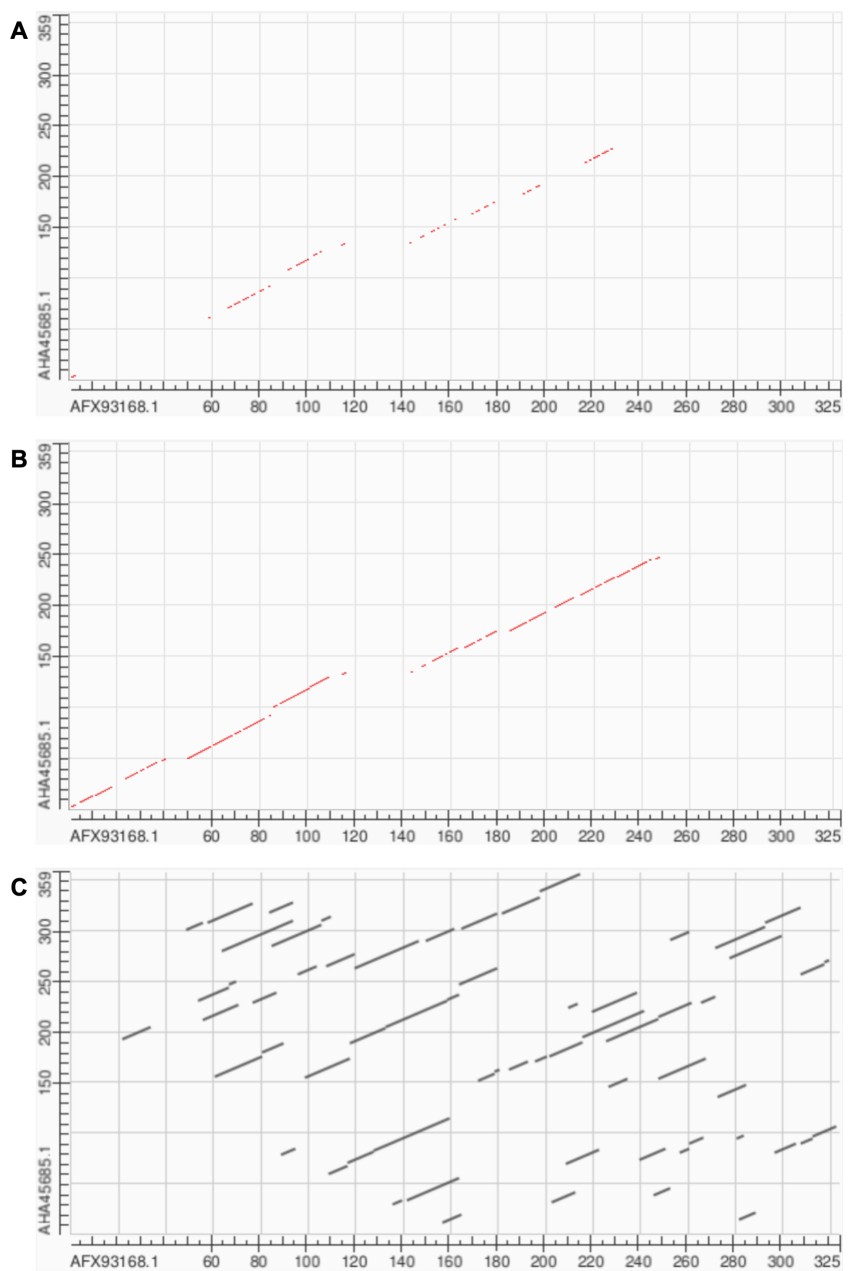

**Figure 6  Soft alignment for protein sequences of AFX93168.1 from Megavirus courdo11 *(Megavirus)* and AHA45685.1 from Hirudovirus strain Sangsue *(Mimivirus)*.** (A) ESM2-3B pre-trained embeddings identify the longest soft alignment path of length 71, whereas (B) ESM2-3B embeddings fine-tuned using the contrastive framework identify the longest path of length 174. (C) Traditional BLASTP homology search generated multiple fragmented regions but was unable to find the longest path.

## Multiple sequence alignments

Vector-clustering MSAs were generated using amino-acid level embeddings of sequences from 100 randomly sampled VOGs from the VOGDB test subset (*McWhite, Armour-Garb*

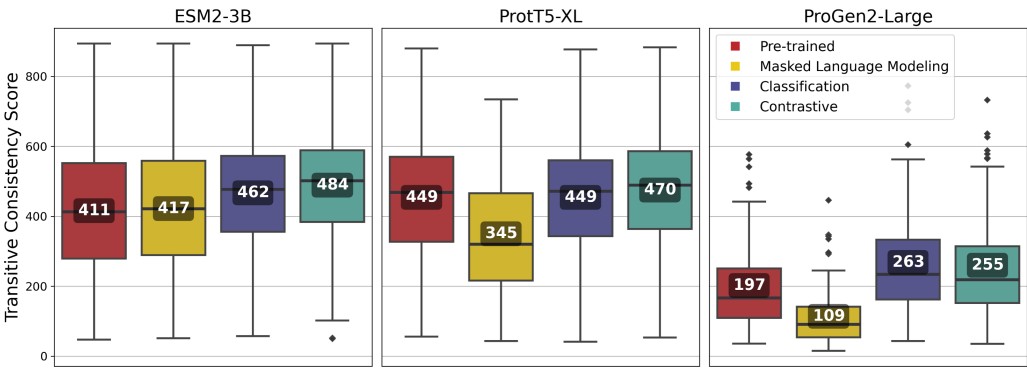

**Figure 7** **Transitive Consistency Scores of 100 vector-clustering MSAs (vcMSAs) annotated with overall mean values.** Each vcMSA contained a maximum of 150 sequences with sequence lengths ranging from 62 to 3,050 amino-acids.

*& Singh, 2023*), and evaluated with the TCS (*Notredame, Higgins & Heringa, 2000*). Fine-tuned models were expected to improve alignment reliability, yielding vcMSAs with higher TCS.

Fine-tuning using the contrastive approach outperformed most other approaches for all of the pLMs, and among the pLMs, ESM2-3B produced the highest-quality vcMSAs with a mean TCS of 484 (Fig. 7).

The impact of pLM fine-tuning on vcMSAs was examined using embeddings for 47 bacteriophage proteins (UvsW helicase), with lengths ranging from 268 to 1,019 residues produced by ESM2-3B fine-tuned using the contrastive approach. The vcMSA generated from embeddings using pre-trained ESM2-3B yielded a TCS of 939, an average Mutual Information (MI) of 0.16, and an average occupancy of 0.51 (Fig. 8A). In contrast, alignments using embeddings from the contrastive fine-tuned ESM2-3B model resulted in a TCS of 972, average MI of 0.22, and occupancy of 0.59 (Fig. 8B). Most notably, pronounced gaps occurred in alignments from the pre-trained model (Fig. 8A). For instance, alanine residues (A) at alignment positions 28–31 and 34–37 remained unaligned due to vcMSA's assessment that their embeddings generated from the pre-trained ESM2-3B model were too distant to indicate homology. Similarly, the arginine (R) and methionine (M) at alignment positions 39–41 remained unaligned, despite multiple instances of each at these positions. Conversely, alignments produced using embeddings from contrastive fine-tuned ESM2-3B produced no gaps in these positions (Fig. 8B). This result agreed with HMM models associated with the protein family Pfam PF00271, which have co-occurrence probabilities of 0.086 and 0.068, respectively, for these positions. Additionally, the contrastive fine-tuned ESM2-3B model accurately aligned residues R and M at alignment position 28, which exhibit HMM probabilities of 0.040 and 0.011, respectively. These alignments were consistent with results obtained from EMBOSS Needle pairwise alignments and Clustal Omega MSAs (data not shown), further substantiating the reliability of vcMSA generated from embeddings obtained using the the contrastive fine-tuned ESM2-3B model.

Peer⌋

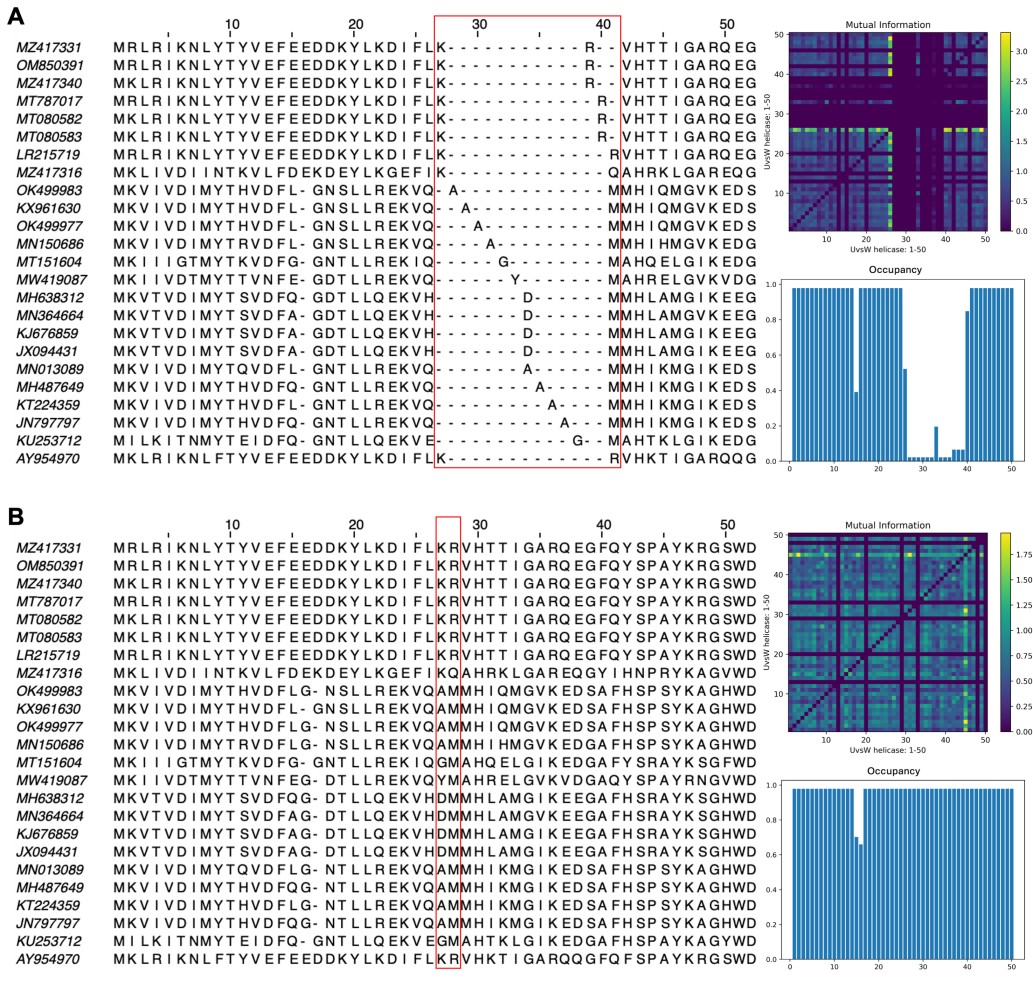

**Figure 8   Subset of the first 50 positions in the vector-clustering MSA of UvsW helicase bacteriophage proteins using (A) pre-trained ESM2-3B and (B) contrastive fine-tuned ESM2-3B.** The pre-trained model aligned sequences with lower mean Mutual Information (MI) (0.16) (top right), occupancy (0.51) (bottom right), and TCS (939). The contrastive fine-tuned ESM2-3B model produced an alignment of greater quality with higher mean MI (0.22), occupancy (0.59), and TCS (972).

## DISCUSSION

Our results demonstrate that fine-tuning substantially enhances the performance of pLMs on tasks critical to understanding viral function. The magnitude of improvement varies depending on the task, training objective, and model architecture. Notably, the contrastive fine-tuned ESM2-3B model consistently outperformed its pre-trained counterpart across pairwise sequence embedding comparisons, clustering, and alignment-based evaluations.

In pairwise embedding assessments, the broadening of cosine similarity distributions, and the improved discrimination between homologous and non-homologous sites suggests that fine-tuning allows the pLM to better represent subtle differences in sequence context. This is in line with previous work indicating that domain-specific fine-tuning can mitigate biases arising from imbalanced training datasets and enhance the model's sensitivity

to biologically relevant variations. Such improvements are particularly crucial for viral proteins, which have historically been underrepresented in the training data for many pLMs.

Clustering experiments further reinforced these findings. The high silhouette scores achieved by the contrastive fine-tuned ESM2-3B illustrate its enhanced ability to group proteins into biologically coherent clusters. For instance, the refined separation of helicase families—where distinct functional subgroups were more clearly delineated compared to pre-trained embeddings—underscores the potential of fine-tuning to preserve and even enhance functionally relevant features within protein families.

Alignment-based assessments provide additional evidence of the benefits conferred by fine-tuning. The contrastively fine-tuned model not only produced longer homologous residue paths in soft alignments but also generated MSAs with higher TCS and improved MI metrics. These improvements suggest that fine-tuned embeddings can capture evolutionary signals more effectively, which is essential for accurate functional annotation and comparative sequence analysis.

Interestingly, while ESM2-3B and, to a moderate extent, ProtT5-XL exhibited clear improvements following fine-tuning, the ProGen2-Large model showed limited gains. As evident through the clustering task, ProGen2 struggled to form coherent clusters, consistently yielding poor silhouette scores under optimal settings. This divergence indicates that the efficacy of fine-tuning may depend on the underlying model architecture and its pre-training regime.

We observed that MLM fine-tuning did not improve model performance on most downstream tasks, likely due to overfitting. As these models were initially trained with MLM (ESM2, ProtT5) or next-token prediction (ProGen2), our results suggest diminishing returns when similar training objectives are applied repeatedly.

Beyond the immediate performance improvements, our study underscores the broader utility of parameter-efficient fine-tuning approaches like LoRA. By dramatically reducing computational requirements, LoRA makes it feasible for research groups with limited resources to adapt large-scale pLMs for specialized applications, such as the study of viral proteins. These advances offer innovative and unprecedented opportunities for understanding viral function and ecology.

## CONCLUSION

Twelve protein language model variants were evaluated for their ability to capture viral protein biology. In these evaluations, the pre-trained pLMs were compared with versions fine-tuned for viral sequences using one of three learning frameworks. Our application of PEFT revealed clear benefits: Using viral proteins to selectively update pLM weights (initially learned through MLM or autoregressive causal language modeling) with classification or contrastive learning frameworks resulted in performance improvements across diverse tasks. Our fine-tuned models demonstrated improved clustering capabilities, with both contrastive ESM2-3B and ProtT5-XL variants producing clearer distinctions among viral protein orthologous groups, families, and domains, indicating that fine-tuning enhances

the models' ability to differentiate distinct protein groups despite low sequence similarity, thus supporting biologically meaningful hierarchies.

Additionally, the fine-tuned models produced better quality alignments, with both contrastive and classification fine-tuning frameworks significantly improving global alignment accuracy and MSA fidelity. These improvements underscore the fine-tuned pLMs' refined ability to capture conserved motifs and infer evolutionary relationships with greater precision. The results were particularly significant for contrastive fine-tuned ESM2-3B, which consistently outperformed all other models and fine-tuned variants. This model showed robust performance across numerous downstream tasks.

We experimented with various learning frameworks, highlighting the benefits of diversifying learning methodologies in fine-tuning pLMs. This diversity in training techniques proved advantageous, as it enriched model representations and enabled superior performance across clustering and alignment tasks. Our findings emphasize that PEFT methods can substantially reduce the computational and memory footprint of training, making these techniques accessible for large models without sacrificing efficiency. In sum, these findings pave the way for more accessible, high-performance tools in bioinformatics, advancing our ability to analyze and interpret complex biological data at scale.

### Funding

This work was supported by National Science Foundation grant numbers 1736030 and 2025567. Access to high performance computational resources, storage, and expertise were provided by the University of Delaware Bioinformatics Data Science Core Facility (RRID:SCR_017696) with support from Delaware INBRE (NIH P20GM103446), the NIH Shared Instrumentation Grant (NIH S10OD028725), the Delaware Biotechnology Institute, and the State of Delaware. The funders had no role in study design, data collection and analysis, decision to publish, or preparation of the manuscript.

### Grant Disclosures

The following grant information was disclosed by the authors:
National Science Foundation: 1736030, 2025567.
Delaware INBRE: NIH P20GM103446.
NIH Shared Instrumentation: NIH S10OD028725.
Delaware Biotechnology Institute.
State of Delaware.

### Competing Interests

The authors declare there are no competing interests.

### Author Contributions

- Rajan Sawhney conceived and designed the experiments, performed the experiments, analyzed the data, prepared figures and/or tables, authored or reviewed drafts of the article, and approved the final draft.

- Barbra D. Ferrell conceived and designed the experiments, analyzed the data, authored or reviewed drafts of the article, and approved the final draft.
- Thibaut Dejean conceived and designed the experiments, performed the experiments, analyzed the data, prepared figures and/or tables, authored or reviewed drafts of the article, and approved the final draft.
- Zachary Schreiber conceived and designed the experiments, performed the experiments, analyzed the data, prepared figures and/or tables, authored or reviewed drafts of the article, and approved the final draft.
- William Harrigan conceived and designed the experiments, analyzed the data, authored or reviewed drafts of the article, and approved the final draft.
- Shawn W. Polson analyzed the data, authored or reviewed drafts of the article, and approved the final draft.
- K. Eric Wommack analyzed the data, authored or reviewed drafts of the article, and approved the final draft.
- Mahdi Belcaid conceived and designed the experiments, analyzed the data, authored or reviewed drafts of the article, and approved the final draft.

### Data Availability

Source code and datasets are available at GitHub and Zenodo: Available at https://github.com/Hawaii-Bioinformatics/ViralFineTuning.

Sawhney, R. (2025). Hawaii-Bioinformatics/ViralFineTuning. Zenodo. Available at https://doi.org/10.5281/zenodo.15865074.

The HeliBase (DNA helicase) dataset is also available at GitHub and Zenodo: Available at https://github.com/ud-veil/HeliBase.

ferrellb. (2025). ud-veil/HeliBase: Initial Release (0.1.0). Zenodo. Available at https://doi.org/10.5281/zenodo.15855844.

### Supplemental Information

Supplemental information for this article can be found online at http://dx.doi.org/10.7717/peerj.19919#supplemental-information.

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
