# Peer review of "Fine-tuning protein language models unlocks the potential of underrepresented viral proteomes"

_PeerJ, doi:10.7717/peerj.19919_

## Round 0.1 · original submission · Minor Revisions

Two experts in the field reviewed your manuscript. Although both recommend minor revisions, it may take some time to address all of them. Please read their comments carefully and revise the manuscript accordingly. Explain why, when you think some of their comments are inappropriate to follow.

Reviewer 1 ·

Basic reporting

The article is generally presented clearly both in text and figures.

The overall coverage of the machine learning and protein language model (PLM) background is thorough. However, there are other related studies on relevant topics such as fine-tuning PLMs for viruses, annotating viral proteins, speciality PLMs for viral proteins, benchmarking PLMs on viral protein mutation prediction, etc.: Flamholz 2024 doi:10.1038/s41564-023-01584-8, Martin 2024 doi:10.1101/2024.07.26.605391, Marathe 2024 arXiv:2412.16262, Rancati 2024 doi:10.1101/2024.12.10.627777, Yu 2025 doi:10.1103/PhysRevResearch.7.013229, Gurev 2025 https://openreview.net/pdf?id=DvC6VL7TJK. It is not necessary to cite and discuss this specific set of papers, but there is virology-specific research involving PLMs that would place this study in a broader context.

The notation in line 243 could be improved, as subscripts i and j are used for both position and sequence indices. My initial reading gave the impression that the "for all i \neq j" referred to multiple positions, not all pairs of sequences.

In Figure 3, the highest score is not as relevant as the median score. The descriptions of these 1000 iterations in the text are unclear. If an iteration refers to the random sampling of sequences from the 500 VOGs, that could be restated to emphasize that the 1000 refers to different random samples. It was also not well-motivated why only 500 VOGs were used from the test set instead of all the test set VOGs in this process.

The mean path lengths reported in Figure 5 appear to be larger than the values in the distributions in some cases.

Experimental design

The research is well within the journal's scope, and the research question addresses a knowledge gap related to the underrepresentation of viral sequences in most PLMs.

In line 243, the methodology to select a pair of sites does not consider 3D contacts, only 1D positions. The sites i and j could be spatially close.

Using the 2D UMAP embedding vectors instead of the full protein embedding vectors for the helicase embeddings is not well motivated. The UMAP vectors will distort the information in the original embedding vectors.

The Contrastive Learning loss function is not written out in full.

In Methods Section 1.2, there are initially 300 VOGs, but then 51 VOGs at the end. If this reduction is due to conservation patterns, that could be stated.

Validity of the findings

Some of the results are not explained well. The MLM objective often fails to improve the pretrained model, and this result is not explored. The details of the fine-tuning processes could provide information about whether this is due to the objective or a problem with the model training (e.g., poor hyperparameters). The CON objective works best for ESM2 and ProtT5 in most scenarios, but hurts ProGen2 clustering results. That is not explained either. Once again, did the fine-tuning not work well, or is it something else?

The opening sentence of the Discussion, "that fine-tuning substantially enhances the performance of pLMs," is too strong. The performance was dependent on the task, objective function, and PLM.

Additional comments

The stratified sampling procedure is not motivated. This creates test splits in which the same VOGs were observed during training. That choice is not wrong, but it requires the fine-tuned PLMs to generalize less than if the splits were based on VOGs with some VOGs held out for testing. That splitting strategy could be touched on in the Discussion because it will affect how well the tune PLMs can generalize to new VOGs.

The GitHub and Hugging Face repositories look good overall. ProGen2 is missing the MLM code. The code is sufficient to provide details about the study and methods, but not rerun all the computational methods because file paths are hard-coded and the VOGDB data is not present in the expected format.

·

Basic reporting

General comments

The scientific approach employed is rational and is supported by multiple similar studies applied to different domains in the AI space. The novelty of this article, therefore, lies in the application of this concept to improving protein search tasks that involve viruses, and the innovative approach used to compare their results with controls.

The quality of English language used in the manuscript is generally high, and it was an engaging read. The introduction and background work are suitably deep and topical, with appropriate referencing. The work would appear to conform to PeerJ standards, and both source code and raw data are supplied as per journal policy. Figures are suitably informative, though image quality is insufficient for some. Additional work should be conducted to improve the interpretability of results (in particular, some basic statistical analysis) and ease of reproducing the authors’ experiments.

Specific comments
Source code. The instructions in both the manuscript and repository README are not sufficient for users to reliably reproduce the authors’ work, whereas it is stated in line 118 that the repository is included for this purpose. Specific things the authors could address include a pip requirements file, conda build file or Docker container, and annotation of which files are used and how to use them (e.g. some of the python and shell files in the subdirectories seem to reference hard-coded location on the authors’ machines – presumably these aren’t required?).

Ln 49. Incorrect direction of double quotation on “grammar”. (On LaTeX, use ``)
Fig. 1. Bit blurry, so might not scale well in print. Looks like PowerPoint – suggest output as a vector image if possible.
Fig. 1 / Ln 197 / Ln 204. Re. “weights are added together” / “LoRA matrices were added to…”. Are they really added, as in matrix addition? Or are they merged (if I remember, the Huggingface method is indeed called add weighted adapter, but there’s a range of strategies – clarity here would be very helpful).
Table 2. Re. “appended subscripts…”. Doesn’t start with a capital; why is this not in the legend?
Ln 220. Were any other metrics considered, e.g., Jaccard? This is just a point of interest, rather than a criticism.
Results. Bar charts with cosine similarity, silhouette scores, longest homologous residue results, and transitive consistency scores (Fig. 2, 3, 5, 7) would greatly benefit from some statistics, to help readers compare and contrast results that look superficially similar, i.e. using asterisks overlaid onto the box plots to denote statistical significance. Specifically, comparing the midpoint (or distribution, if authors prefer) of results in fine-tuning experiments to control observations.
Fig. 6. Also looks a bit blurry.

Richard Mayne, senior bioinformatician in my group, conducted this review, and I concur with the comments and opinions expressed

Experimental design

The manuscript examines a proposed partial solution to the current problem of the underrepresentation of viral proteins in protein large language model training sets. The authors use a process of transfer learning called fine-tuning to improve the performance of pre-trained models, which involves updating model parameters based on a small additional dataset. The authors applied a variety of fine-tuning techniques to three popular models according to a scheme called LoRA, using large publicly available virus protein databases, and used three sets of metrics for comparing results with those from unmodified control models. Findings presented indicate that fine-tuning helps support a variety of downstream analyses in some cases, but this is nuanced depending on the fine-tuning technique used and which model it has been applied to.

Validity of the findings

-

---

## Round 0.2 · accepted · Accept

As the handling editor, I confirmed that the authors have addressed all the comments. I believe the current manuscript is now suitable for publication. Congratulations!